# Fuzzy paraphrases in learning word representations with a lexicon

**Yuanzhi Ke & Masafumi Hagiwara**
Department of Information and Computer Science
Keio University
Hiyoshi 3-14-1, Kohokuku, Yokohama City, Kanagawa, Japan
{enshika8811.a6, hagiwara}@keio.jp

## Abstract

A synonym of a polysemous word is usually only the paraphrase of one sense among many. When lexicons are used to improve vector-space word representations, such paraphrases are unreliable and bring noise to the vector-space. The prior works use a coefficient to adjust the overall learning of the lexicons. They regard the paraphrases equally. In this paper, we propose a novel approach that regards the paraphrases diversely to alleviate the adverse effects of polysemy. We annotate each paraphrase with a degree of reliability. The paraphrases are randomly eliminated according to the degrees when our model learns word representations. In this way, our approach drops the unreliable paraphrases, keeping more reliable paraphrases at the same time. The experimental results show that the proposed method improves the word vectors. Our approach is an attempt to address the polysemy problem keeping one vector per word. It makes the approach easier to use than the conventional methods that estimate multiple vectors for a word. Our approach also outperforms the prior works in the experiments.

## 1 Introduction

Vector-space representations of words are reported useful and improve the performance of the machine learning algorithms for many natural language processing tasks such as name entity recognition and chunking (Turian et al., 2010), text classification (Socher et al., 2012; Le & Mikolov, 2014; Kim, 2014; Joulin et al., 2016), topic extraction (Das et al., 2015; Li et al., 2016), and machine translation (Zaremba et al., 2014; Sutskever et al., 2014).

People are still trying to improve the vector-space representations for words. Bojanowski et al. (2016) attempt to improve word vectors by involving character level information. Other works (Yu & Dredze, 2014; Xu et al., 2014; Faruqui et al., 2015; Bollegala et al., 2016) try to estimate better word vectors by using a lexicon or ontology. The idea is simple: because a lexicon or ontology contains well-defined relations about words, we can use them to improve word vectors.

However, for a polysemous word, one of its synonym does not always mean the same thing with the original one under different contexts. For example, the word "point" equals "score" in "Team A got 3 points", but does not in "my point of view." A method to address this issue is to estimate a vector for each word sense (Huang et al., 2012; Chen et al., 2014) or per word type (Neelakantan et al., 2014). However, it requires additional word sense disambiguation or part-of-speech tagging to use such word vectors.

In this paper, we propose a method to improve the vector-space representations using a lexicon and alleviate the adverse effect of polysemy, keeping one vector per word. We estimate the degree of reliability for each paraphrase in the lexicon and eliminate the ones with lower degrees in learning. The experimental results show that the proposed method is effective and outperforms the prior works. The major contributions of our work include:

- We propose a novel approach involving fuzzy sets to reduce the noise brought by polysemous words in the word vector space when a lexicon is used for learning, and a model to use the fuzzy paraphrase sets to learn the word vector space.

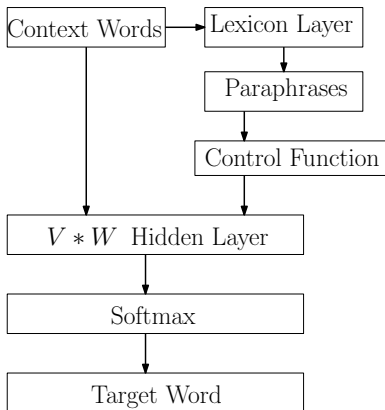

Figure 1: The process flow of the proposed method.

- Although some prior works propose to solve the polysemy problem by estimating one vector per word sense or type, using such word vectors requires additional pre-process. Our proposed method keeps one vector per word. It makes the word vectors easier to use in practical terms: it is neither necessary to disambiguate the word senses nor to tag the part-of-speeches before we use the word vectors.

We give an introduction of our proposed method in section 2. We show the effects of different paraphrase sets, parameters, corpus size, and evaluate the effectiveness of our approach by comparing to simpler algorithms in section 3. We compare our approach with the prior works via an evaluation experiment in section 4. We give the findings, conclusions and outlook in section 5.

## 2 THE PROPOSED METHOD

### 2.1 FUZZY PARAPHRASES

As described in section 1, whether a polysemous word's paraphrase is the same as the original depends on the context.

Henceforth, if we simply use all the paraphrases of a word in the lexicon to improve the word vector without discrimination, they may sometimes bring noise to the vector-space.

A conventional method for them is to give each word sense a vector. However, such vector-spaces require additional word sense disambiguation in practical use.

Here, we propose a method to alleviate the adverse effects of polysemous words' paraphrases without word sense disambiguation. Our idea is to annotate each paraphrase with a degree about its reliability, like a member of a fuzzy set. We call such paraphrases as "fuzzy paraphrases", and their degrees as the "memberships."

### 2.2 LEARNING WITH FUZZY PARAPHRASES

We also propose a novel method to jointly learn corpus with a lexicon, in order to use fuzzy paraphrases to improve the word vectors.

If the meanings of two words are totally the same, they can replace each other in a text without changing the semantic features. Henceforth, we can learn the lexicon by replacing the words in the corpus with its lexical paraphrases.

We learn the word vectors by maximizing the probability of a word for a given context, and also for a generated context where words are replaced by their paraphrases randomly. The memberships of the fuzzy paraphrases are used here to control the probability that the replacements occur by a control function as shown in Figure 1.

For a text corpus $T$, denote $w_i$ the $i$th word in $T$, $c$ the context window, $w_j$ a word in the context window, $L_{w_j}$ the paraphrase set of $w_j$ in the lexicon $L$, $w_k$ the $k$th fuzzy paraphrase in $L_{w_j}$, and $x_{jk}$ the membership of $w_k$ for $w_j$, the objective is

$$\sum_{w_i \in T} \sum_{(i-c) \leq j \leq (i+c)} \left[ \log p(w_i|w_j) + \sum_{w_k \in L_{w_j}}^{L_{w_j}} f(x_{jk}) \log p(w_i|w_k) \right]. \tag{1}$$

The function $f(x_{jk})$ of the membership $x_{jk}$ is a specified drop-out function. It returns 0 more for the paraphrases that have lower memberships, and 1 more for the others.

## 2.3 MEMBERSHIP ESTIMATION & CONTROL FUNCTION $f(x)$

Looking for a control function that is easy to train, we notice that if two words are more often to be translated to the same word in another language, the replacement of them are less likely to change the meaning of the original sentence. Thus, we use a function of the bilingual similarity (denoted as $S_{jk}$) as the membership function:

$$x_{jk} = g(S_{jk}). \tag{2}$$

There have been works about calculating the similarity of words using such bilingual information. A lexicon called the paraphrase database (PPDB) provides scores of the similarity of paraphrases on the basis of bilingual features (Ganitkevitch et al., 2013; Pavlick et al., 2015b;a).

We scale the similarity score of the paraphrase $w_k$ to $[0, 1]$ in PPDB2.0 as the memberships, and draw the values of $f(x_{jk})$ from a Bernoulli distribution subjected to them. Denote $S_{jk}$ the similarity score of word $w_j$ and $w_k$ in PPDB2.0, the value of $f(x_{jk})$ is drawn from the Bernoulli distribution:

$$f(x_{jk}) \sim Bernoulli(x_{jk}), \tag{3}$$

$$x_{jk} = \frac{S_{jk}}{\max_{j \in T, k \in L} S_{jk}}. \tag{4}$$

## 2.4 TRAINING

We do not need to train $f(x_{jk})$ using the method described above. The model can be trained by negative sampling (Mikolov et al., 2013b): For word $w_O$ and a word $w_I$ in its context, denote $A_I$ as the set of the paraphrases for $w_I$ accepted by $f(x_{jk})$, we maximize $\log p(w_O|w_I)$ by distinguishing the noise words from a noise distribution $P_n(w)$ from $w_O$ and its accepted paraphrases in $A_I$ by logistic regression:

$$\log p(w_O|w_I) = \log \sigma(v_{w_O}{}^{\mathrm{T}} v_{w_I}) + \sum_{i=1}^{n} E_{w_i} \sim P_n(w)[\log \sigma(-v_{w_i}{}^{\mathrm{T}} v_{w_I})], w_i \neq w_O, w_i \notin A_I \tag{5}$$

Here, $v_{w_O}{}^{\mathrm{T}}$ and $v_{w_i}{}^{\mathrm{T}}$ stand for the transposed matrices of $v_{w_O}$ and $v_{w_i}$, respectively. $n$ is the number of negative samples used. $\sigma(x)$ is a sigmoid function, $\sigma(x) = 1/(1 + e^{-x})$.

## 3 MODEL EXPLORATION

### 3.1 CORPUS FOR EXPERIMENTS

We use enwiki9[1] mainly for tuning and model exploration. It has a balanced size(1 GB), containing 123,353,508 tokens. It provides enough data to alleviate randomness while it does not take too much time for our model to learn.

---

[1]http://mattmahoney.net/dc/enwiki9.zip

Table 1: The results of 10 times repeated learning and test under each benchmark. The vector-space dimension is set to 100. Enwiki9 is used as the corpus. The maximum, minimum, and the margin of error are marked bold.

| Benchmark | SimLex | WS353 | RW | MEN | SEM | SYN |
|---|---|---|---|---|---|---|
| 1 | 29.41 | 62.02 | 38.12 | 60.00 | 13.26 | **27.77** |
| 2 | **29.57** | 62.49 | 38.26 | 60.39 | **12.70** | 27.27 |
| 3 | 29.48 | **61.04** | 39.90 | 59.80 | 13.89 | 26.94 |
| 4 | 29.52 | 60.20 | 39.68 | 59.81 | **14.02** | 27.11 |
| 5 | 28.69 | **63.45** | 38.65 | 60.16 | 12.94 | 26.87 |
| 6 | 29.26 | 61.95 | 39.13 | 59.73 | 13.75 | 26.60 |
| 7 | 29.46 | 62.90 | 39.12 | **60.45** | 13.42 | **25.98** |
| 8 | 28.51 | 62.96 | **37.93** | **59.31** | 13.58 | 27.10 |
| 9 | 29.13 | 62.44 | **39.91** | 59.75 | 13.98 | 26.89 |
| 10 | **28.59** | 60.66 | 38.67 | 60.24 | 13.66 | 26.98 |
| **Margin of Error** | **0.98** | **2.41** | **1.98** | **1.14** | **1.32** | **1.79** |

We use ukWaC (Baroni et al., 2009) to compare with the prior works in section 4. But we do not use it for model exploration, because it takes more than 20 hours to learn it, as an enormous corpus containing 12 GB text.

## 3.2 BENCHMARKS

We used several benchmarks. They are Wordsim-353 (`WS353`) (Finkelstein et al., 2001) (353 word pairs), SimLex-999 (`SimLex`) (Hill et al., 2016) (999 word pairs), the Stanford Rare Word Similarity Dataset (`RW`) (Luong et al., 2013) (2034 word pairs), the MEN Dataset (`MEN`) (Bruni et al., 2014) (3000 word pairs), and the Mikolov's (Google's) word analogical reasoning task (Mikolov et al., 2013a).

`WS353`, `SimLex`, and `RW` are gold standards. They provide the similarity of words labeled by humans. We report the Spearman's rank correlation ($\rho$) for them.

Mikolov's word analogical reasoning task is another widely used benchmark for word vectors. It contains a semantic part (`SEM`), and a syntactic part (`SYN`). We use the basic way suggested in their paper to find the answer for it: to guess word $b'$ related to $b$ in the way how $a'$ is related to $a$, the word closest in cosine similarity to $a' - a + b$ is returned as $b'$.

We find that the benchmark scores change every time we learn the corpus, even under the same settings. It is because that the models involve random numbers. Therefore we should consider the margin of error of the changes when we use the benchmarks.

To test the margin of error, we firstly used our proposed method to repeat learning enwiki9 for 10 times under the same parameters. Then we tested the vectors under each benchmark, to find the margin of error. In each test, we used the same parameters: the vector dimension was set to 100 for speed, the window size was set to 8, and 25 negative samples were used. The results are shown in Table 1. We use them to analyze the other experimental results later.

## 3.3 DIFFERENT TYPES OF PARAPHRASES

In PPDB2.0, there are six relationships for paraphrases. For word $X$ and $Y$, the different relationships between them defined in PPDB2.0 are shown in Table 2. We do not consider the exclusion and independent relations because they are not semantic paraphrases. Those of equivalence are the most reliable because they are the closest ones. But we still want to know whether it is better to take

Table 2: Different types of relationships of paraphrases in PPDB2.0(Pavlick et al., 2015b;a).

| Relationship Type | Description |
|---|---|
| Equivalence | $X$ is the same as $Y$ |
| Forward Entailment | $X$ is more specific than/is a type of $Y$ |
| Reverse Entailment | $X$ is more general than/encompasses $Y$ |
| Exclusion | $X$ is the opposite of $Y$ / $X$ is mutually exclusive with $Y$ |
| OtherRelated | $X$ is related in some other way to $Y$ |
| Independent | $X$ is not related to $Y$ |

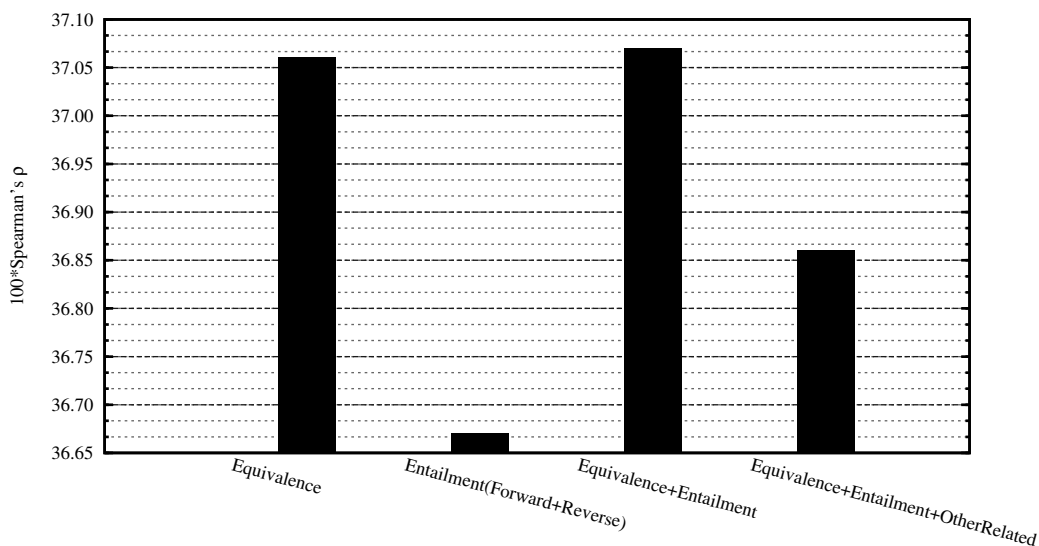

Figure 2: The $\rho$ for `SimLex` using different paraphrase sets. The corpus is enwiki9. The vector-space dimension is set to 300. The context window size is set to 8. 25 negative samples are used in learning.

the entailment and the other related paraphrases into consideration. We learn enwiki9 with different paraphrase sets and use `SimLex` to evaluate the trained vectors.

Figure 2 compares the performance using different paraphrase sets, tested by `SimLex`. We can see that it is best to use the equivalence and entailment (forward + reverse) paraphrases together or use only the equivalence paraphrases. Only using the entailment paraphrases is weak. Involving the other related paraphrases deteriorates the performance. We use the Equivalence and Entailment paraphrases in the experiments according to these results.

## 3.4 Effects of Parameters

We use our proposed method to learn enwiki9 under different parameter settings to evaluate the effects of parameters. We firstly learn enwiki9 under different parameter settings and then test the vectors using `SimLex`, `WS353`, `RW`, `MEN`, `SEM` and `SYN`. We report Spearman's rank correlation $\rho$ for `SimLex`, `WS353`, `RW` and `MEN`, the percentage of correct answers for `SEM` and `SYN`.

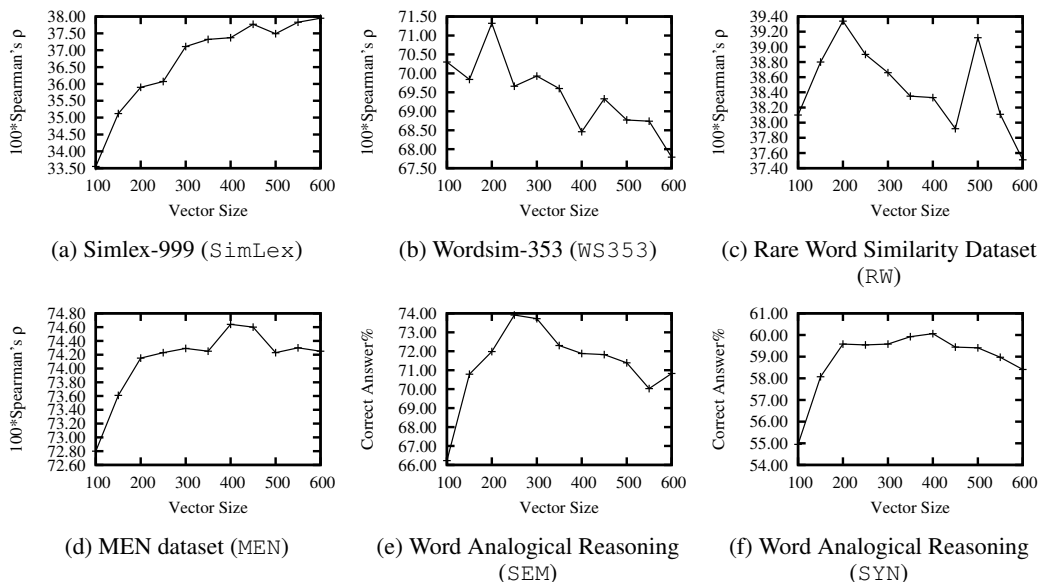

Figure 3: The scores of the benchmarks using different vector-space dimensions. For `WS353`, `SimLex`, `RW` and `MEN`, we report $100 * \rho$ (Spearman's rank correlation). For word analogical reasoning, we report the percentage of the correct answers. The context window size is set to 8. The number of negative samples is set to 25.

### 3.4.1 EFFECTS OF VECTOR SPACE DIMENSION

We compare the benchmarks using different vector-space dimensions. Figure 3 shows the change of each benchmark's scores under different dimensions.

We find that:

- The larger vectors do not bring the better performance for most of the benchmarks (except `SimLex`), although some previous works suggest that the higher dimensions brings better performance for their methods (Pennington et al., 2014; Levy & Goldberg, 2014b).

- The curves of `SimLex` and `SYN` are gradual. However, there are several abrupt changes in the others. And those of `WS353` and `RW` do not change gradually.

- The best dimension for different benchmarks is not consistent.

The differences in the content of the benchmarks may cause the inconsistence. For example, `SimLex` rates related but dissimilar words lower than the other word similarity benchmarks (Hill et al., 2016; Chiu et al., 2016). The results suggest that the best dimensions for our method depends on the task.

### 3.4.2 EFFECTS OF CONTEXT WINDOW SIZE

We compared the benchmarks using different context window sizes. They are shown in Figure 4. Previous works argue that larger window sizes introduce more topic words, and smaller ones emphasize word functions (Turney, 2012; Levy & Goldberg, 2014a; Levy et al., 2015; Hill et al., 2016; Chiu et al., 2016). Different context window sizes provide different balances between relatedness and similarity. The best window size depends on what we want the vectors to be. We also see that in our results.

The relationship between the window size and performance depends on how they rate the pairs. For example, `WS353` rates word pairs according to association rather than similarity (Finkelstein et al., 2001; Hill et al., 2016). As larger window capture relatedness rather than similarity, the results show

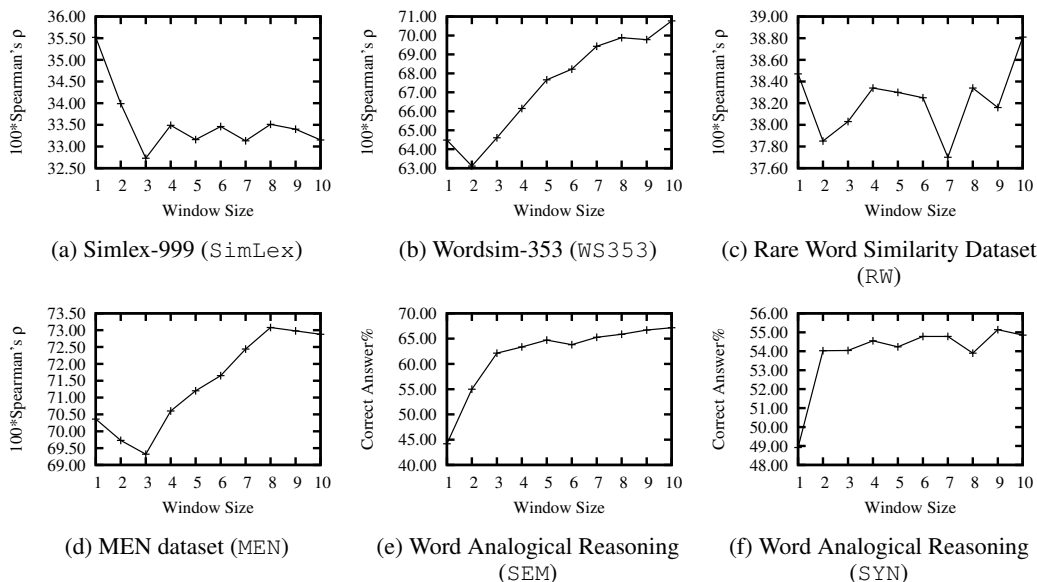

Figure 4: The scores of the benchmarks using different context window sizes. For `WS353`, `SimLex`, `RW` and `MEN`, we report $100 * \rho$ (Spearman's rank correlation). For word analogical reasoning, we report the percentage of the correct answers. We use 100-dimension vectors. The number of negative samples is set to 25.

that the larger the window is, the better for `WS353`. The MEN dataset also prefer relatedness than similarity (Bruni et al., 2014), but they gave annotators examples involving similarity[2]. It may be the reason that the windows larger than 8 deteriorate the benchmarks based on `MEN` (Figure 4d). The standards of `WS353` and `MEN` to rate the words are similar (Bruni et al., 2014). It leads to their similar curves (Figure 4b and 4d). The worst window sizes of them are also close. When the window size is set to about 2 or 3, respectively, the balance of similarity and relatedness is the worst for them.

Unlike the other word similarity dataset, `SimLex` rates synonyms high and related dissimilar word pairs low. Therefore, the smallest window is the most suitable for `SimLex` because it is best for capturing the functional similarity.

The results of `RW` differs from the others (Figure 4c). There are many abrupt changes. The best window size is 10, but 1 is better than 2-9. The dataset contains rare words. Because of their low frequencies, usage of broad context window may be better to draw features for them. However, additional words introduced by larger windows may also deteriorate the vectors of unusual words. For such tasks requiring rare word vectors of high quality, we should be careful in tuning the context window size.

For Google's word analogical tasks (`SEM` and `SYN`), the questions are quite related to the topic or domain. For examples, there are questions about the capitals of the countries. They are associated but not synonymous. Therefore a larger window is usually better. However for `SYN`, using window size 9 is a little better than 10 in Figure 4d and for `MEN` 8 is best in Figure 4f. It may be because that if the window is too large, it introduces too many words and reduces the sparsity (Chiu et al., 2016).

We can consider that the best context window size depends on the task, but we should avoid using too large window.

### 3.4.3 EFFECTS OF NEGATIVE SAMPLES

We also explored the effects of the number of negative samples. The results are shown in Figure 5.

---

[2]According to their homepage: http://clic.cimec.unitn.it/ elia.bruni/MEN.html.

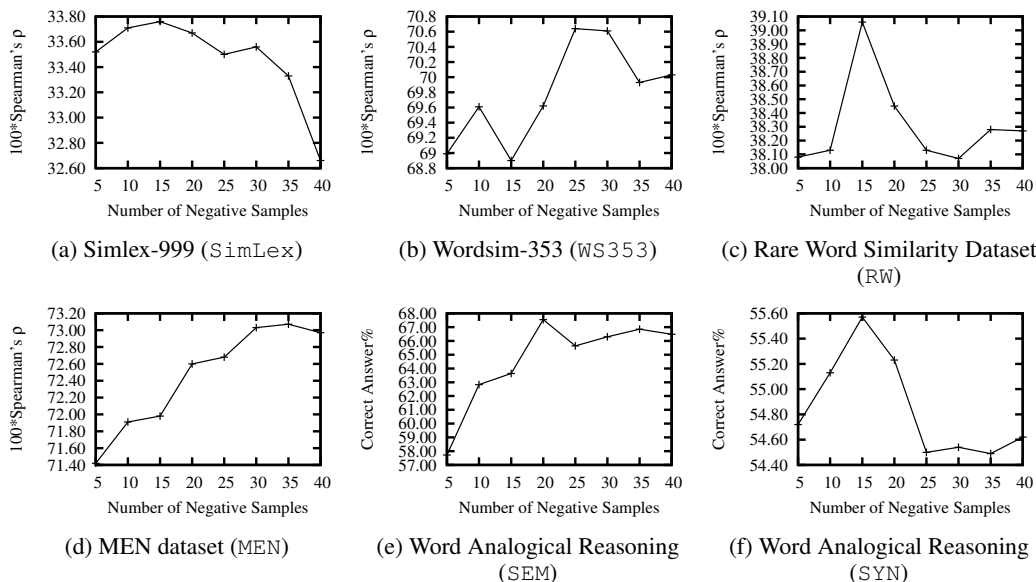

(a) Simlex-999 (`SimLex`) (b) Wordsim-353 (`WS353`) (c) Rare Word Similarity Dataset (`RW`)

(d) MEN dataset (`MEN`) (e) Word Analogical Reasoning (`SEM`) (f) Word Analogical Reasoning (`SYN`)

Figure 5: The scores of the benchmarks using different numbers of negative samples. For `WS353`, `SimLex`, `RW` and `MEN`, we report $100 * \rho$ (Spearman's rank correlation). For word analogical reasoning, we report the percentage of the correct answers. We use 100-dimension vectors. The context window size is set to $8$.

In Figures 5a, 5c and 5f, we see that overfitting occurs when we use more than 15 negative samples. In Figure 5b and Figure 5e, it occurs from 25 and 20, respectively. In Figure 5d, the performance does not change very much when we use more than 30 negative samples.

The results indicate that too many negative samples may cause overfitting. For 3 of the 6 benchmarks, it is best to use 15 negative samples. But we should be careful in practice use because the other different results suggest that the best number depends on the task.

The abrupt change at around 15 in Figure 5b is interesting. `WS353` is the smallest dataset among those we used. Because of the small size, the effects of randomness may cause such singularities when the vector-space is not well trained.

## 3.5 EFFECTS OF THE CONTROL FUNCTION & THE CORPUS SIZE

In this section, we evaluate the effectiveness of our fuzzy approach, by comparing to the situations that set $f(x)$ in Equation (1) as:

- $f(x) = 1$: It makes the model regard all paraphrases equally. They are all used without drop-out.
- $f(x) = 0$: It makes the model use no paraphrases, equivalent to CBOW.

It is also a good way to show the effects of corpus size by comparing the proposed method to the situations above using corpora in varying size. Therefore we discuss them together in this section.

We use text8[3] together with eEnwiki9 and ukWaC described in section 3.1. It is a small corpus containing 100 MB text. To show the difference, we report the benchmarks scores including not only `SimLex`, but also `MEN`, and the word analogical task (`SEM` and `SYN`). They are the other benchmarks that are shown relatively solid in section 3.2. The vector-space dimension is set to 300. The context window size is set to $8$. 25 negative samples are used in learning. The results are shown in Figure 6.

---

[3]http://mattmahoney.net/dc/text8.zip

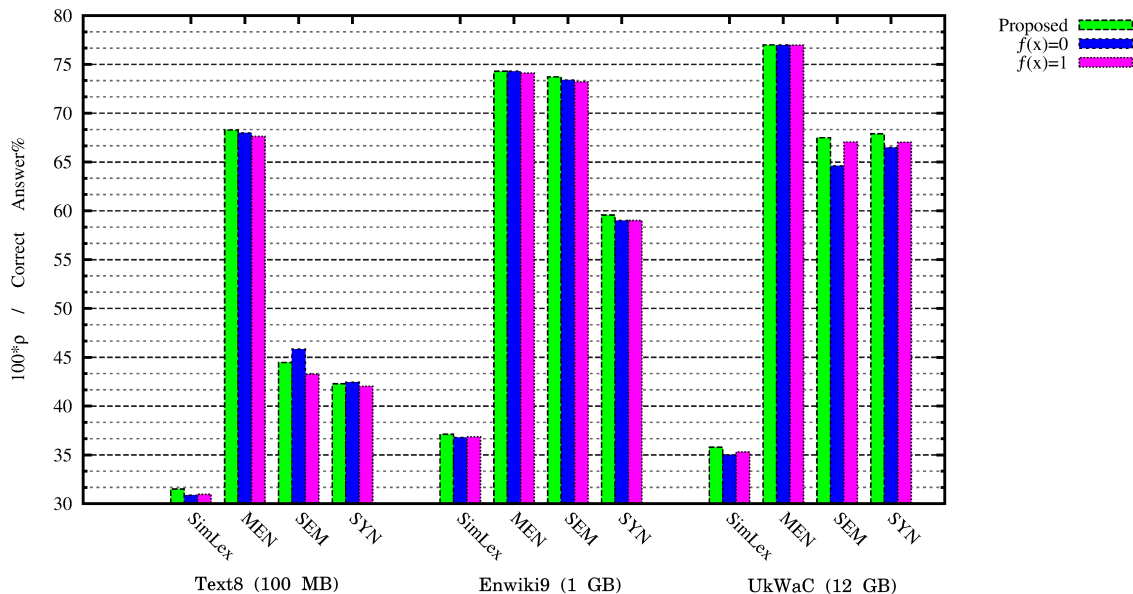

Figure 6: The comparison of using the proposed function described in section 2.3, $f(x) = 0$ (equivalent to CBOW) and $f(x) = 1$ (no drop-out) as the control function. They are compared under different corpora in varying size. The green bar (the left) indicates the scores of the proposed function; the blue bar (the middle) indicates the scores of $f(x) = 0$; the pink bar (the right) indicates the scores of $f(x) = 1$. We report $100 * \rho$ for SimLex and MEN, the percentage of correct answers for SEM and SYN. The vector-space dimension is set to 300. The context window size is set to 8. 25 negative samples are used in learning.

We can see that:

- The proposed function outperforms the others for SimLex and MEN under text8, for all the benchmarks under enwiki9, for SimLex, SEM and SYN under ukWaC.

- The proposed function is always better than $f(x) = 1$ in the experiments, no matter what the benchmark is or how big the corpus is.

- For SEM, the proposed function is weaker than $f(x) = 0$ under text8, slightly better under enwiki9, and obviously outperforms $f(x) = 0$ under ukWaC. As the proposed function outperforms under larger corpora, the relatively low scores under text8 may be caused by the effects of randomness: the proposed function involves random numbers; they bring huge instability under such tiny corpora. Another possible reason is that the control function is less useful for text8 because there are few polysemous words in the tiny corpus.

- There is no advantages to use $f(x) = 1$ instead of $f(x) = 0$ for both text8 and enwiki9. It shows that learning the context words replaced by paraphrases may be not a good idea without fuzzy approaches. However, if we use the proposed control function, the results are better and go beyond those of $f(x) = 0$ in most tests. It shows that the control function utilizing fuzzy paraphrases improves the performance.

Therefore, we can see that the proposed control function using the fuzzy paraphrases annotated with the degrees of reliability improves the quality of the learned word vector-space.

## 4 COMPARISON WITH THE PRIOR WORKS

We compared our work to the prior works using a lexicon to improve word vectors. However, we failed to use the public code to reproduce the works of Yu & Dredze (2014) and Bollegala et al. (2016). We also failed to find an available implementation of Xu et al. (2014). Hence, we use the

Table 3: Comparison to the prior Works. The scores of the prior works under ukWaC are from Bollegala et al. (2016). The `SYN` score of ours and Bollegala's are marked as best together because the margin of error is 1.79 as shown in Table 1.

| Method | MEN | SEM | SYN |
|---|---|---|---|
| Our Proposed Method | **76.99** | **67.48** | **67.89** |
| Bollegala et al. (2016) | 70.90 | 61.46 | **69.33** |
| Yu & Dredze (2014) | 50.10 | - | 29.90 |
| R-Net (Xu et al., 2014) | - | 32.64 | 43.46 |
| C-Net (Xu et al., 2014) | - | 37.07 | 40.06 |
| RC-Net (Xu et al., 2014) | - | 34.36 | 44.42 |
| Faruqui et al. (2015)(Pretrained by CBOW) | 60.50 | 36.65 | 52.50 |
| Faruqui et al. (2015)(Pretrained by Skipgram) | 65.70 | 45.29 | 65.65 |

same corpus and benchmarks with Bollegala et al. (2016) and compare our results with the reported scores of the prior works in their paper. The benchmarks are:

- The MEN Dataset (`MEN`);
- Word Analogical Reasoning Task (`SEM` and `SYN`).

Rubenstein-Goodenough dataset (`RG`) (Rubenstein & Goodenough, 1965) is also used in their works. However, we do not use it, because it fails the sanity check in Batchkarov et al. (2016): $\rho$ may increase when noise is added.

We use ukWaC to learn the word vectors, the same with Bollegala et al. (2016). We also use the same parameters with the prior works: The vector-space dimension is set to 300; the context window size is set to 8; the number of negative samples is set to 25. Then we calculate the cosine similarity of the words and report $100 * \rho$ for `Men`. We use the add method described in section 3.2 and report the percentage of correct answers, for the word analogical reasoning task.

Table 3 shows the results of the experiments. The of `MEN` and `SEM` is 0.86 and 0.44 as shown in Table 1. Therefore we see that our proposed method outperforms the prior works under these benchmarks. We consider our score for `SYN` is as good as Bollegala et al. (2016) achieved, and better than the others, because its margin of error is 1.79 as shown in Table 1.

## 5 CONCLUSION & THE FUTURE WORKS

We proposed a fuzzy approach to control the contamination caused by the polysemous words when a lexicon is used to improve the vector-space word representations. We annotate each paraphrase of a word with a degree of reliability, like the members of a fuzzy set with their memberships, on the basis of their multilingual similarities to the original ones. We use the fuzzy paraphrases to learn a corpus by jointly learning a generated text, in which the original words are randomly replaced by their paraphrases. A paraphrase is less likely to be put into the generated text if it has lower reliability than the others, and vice versa.

We tested the performance using different types of paraphrases in the lexicon PPDB2.0 and find that it is best to use the equivalence type and the entailment type. Using other related paraphrases deteriorates the performance.

We explored the effects of parameters. We find that the best parameter setting depends on the task. We should tune the model carefully in practical use.

We evaluated the effectiveness of our approach by comparing it to the situations that simpler functions are used to control replacements: $f(x) = 1$ which accepts all, and $f(x) = 0$ which rejects

all. We also repeated the experiments under a tiny, a medium sized, and a large corpus, to see the effects of the corpus size on the effectiveness. Our approach achieves the best in 3 of 4 benchmarks under the tiny corpus, and in all benchmarks under the medium sized and the large one. The results indicate that our approach is effective to improve the word vectors.

Our proposed method also achieved the top scores, compared with the prior works.

Unlike the previous works that solve the problems about polysemy by estimating a vector for each word sense or word type, our approach keeps one vector per word. It makes the word vectors easier to use in practical terms: it is neither necessary to disambiguate the word senses nor to tag the part-of-speeches before we use the word vectors.

The fuzzy paraphrases can also be employed for the other models with some changes. We are going to show it in the future. The proposed idea for the polysemy problem without word sense disambiguation is meaningful especially for practical use because it saves the effort of part-of-speech tagging and word sense disambiguation.

Besides, the control function may be more accurate if it considers all the context. We are also going to work on it in the future.

We have opened the source of a demo of the proposed method online[4].

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
