# Peer review of "Fuzzy paraphrases in learning word representations with a lexicon"

_ICLR 2017 — rejected_

[Official Review · AnonReviewer1 · rating 6 · confidence 4 · 16 Dec 2016]
**Interesting idea, unconvincing experiments**

This paper introduces the concept of fuzzy paraphrases to aid in the learning of distributed word representations from a corpus augmented by a lexicon or ontology. Sometimes polysemy is context-dependent, but prior approaches have neglected this fact when incorporating external paraphrase information during learning. The main idea is to introduce a function that essentially judges the context-sensitivity of paraphrase candidates, down-weighting those candidates that depend strongly on context. This function is inferred from bilingual translation agreement.

The main argumentation leading to the model selection is intuitive, and I believe that the inclusion of good paraphrases and the elimination of bad paraphrases during training should in principle improve word representation quality. However, the main questions are how well the proposed method achieves this goal, and, even if it achieves it well, whether it makes much difference in practical terms.

Regarding the first question, I am not entirely convinced that the parameterization of the control function f(x_ij) is optimal. It would have been nice to see some experiments investigating different choices, in particular some baselines where the effect of f is diminished (so that it reduces to f=1 in the limit) would have been interesting. I also feel like there would be a lot to gain from having f be a function of the nearby word embeddings, though this would obvious incur a significant slowdown. (See for example 'Efficient Non-parametric Estimation of Multiple Embeddings per Word in Vector Space' by Neelakantan et al, which should probably be cited.) As it stands, the experimental results do not clearly distinguish the fuzzy paraphrase approach from prior work, i.e. tables 3 and 4 do not show major trends one way or the other.

Regarding the second question, it is hard to draw many conclusions from analogy tasks alone, especially when effects unrelated to good/bad paraphrasing such as corpus size/content, window size, vocabulary size, etc., can have an outsize effect on performance. 

Overall, I think this is a good paper presenting a sensible idea, but I am not convinced by the experiments that the specific approach is achieving its goal. With some improved experiments and analysis, I would wholeheartedly recommend this paper for acceptance; as it stands, I am on the fence.

[Official Review · AnonReviewer3 · rating 3 · confidence 4 · 19 Dec 2016]
**Difficult to understand**

This paper tries to leverage an external lexicon / knowledge base to improve corpus-based word representations by determining (in a fuzzy way) which potential paraphrase is the most appropriate in a particular context.

I think this paper is a bit lost in translation. The grammatical and storytelling styles made it really difficult for me to concentrate, and even unintelligible at times. One of the most important criteria in a conference paper is to communicate one's ideas clearly; unfortunately, I do not feel that this paper meets that standard.

In addition, the evaluation is rather lacking. There are many ways to evaluate word representations, and Google's analogy dataset has many issues (see, for example, Linzen's paper from RepEval 2016, as well as Drozd et al., COLING 2016).

Finally, this work does not provide any qualitative result or motivation. Why does this method work better? Where does it fail? What have we learned about word representations / lexicons / corpus-based methods in general?

[Official Review · AnonReviewer2 · rating 5 · confidence 3 · 20 Dec 2016]
**Good idea but not enough to convince**

This paper proposes a method for estimating the context sensitivity of paraphrases and uses that to inform a word embedding learning model. The main idea and model are presented convincingly and seem plausible. The main weaknesses of the paper are shortcomings in the experimental evaluation and in the model exploration. The evaluation does not convincingly determine whether the model is a significant improvement over simpler methods (particularly those that do not require the paraphrase database!). Likewise, the model section did not convince me that this was the most obvious model formulation to try. The paper would be stronger if model choices were explained more convincingly or - better yet - alternatives were explored.

On balance I lean towards rejecting the paper and encouraging the authors to submit a revised and improved version at a near point in the future.

Detailed/minor points below:

1) While the paper is grammatically mostly correct, it would benefit from revision with the help of a native English speaker. In its current form long sections are very difficult to understand due to the unconventional sentence structure.
2) The tables need better and more descriptive labels.
3) The results are somewhat inconclusive. Particularly in the analogy task in Table 4 it is surprising that CBOW does better on the semantic aspect of the task than your embeddings which are specifically tailored to be good at this?
4) Why was "Enriched CBOW" not included in the analogy task?
5) In the related work section several papers are mentioned that learn embeddings from a combination of lexica and corpora, yet it is repeatedly said that this was the first work of such a kind / that there hasn't been enough work on this. That feels a little misleading.

[Final Decision · Program Chairs · 06 Feb 2017]
**ICLR committee final decision**

The reviewers agree that the paper's clarity and experimental evaluation can be improved.